# Passive Daytime Radiative Cooling of Silica Aerogels

**DOI:** 10.3390/nano13030467

**Published:** 2023-01-24

**Authors:** Bingjie Ma, Yingying Cheng, Peiying Hu, Dan Fang, Jin Wang

**Affiliations:** 1School of Nano-Tech and Nano-Bionics, University of Science and Technology of China, Hefei 230026, China; 2Key Laboratory of Multifunctional Nanomaterials and Smart Systems, Suzhou Institute of Nano-Tech and Nano-Bionics, Chinese Academy of Sciences, Suzhou 215123, China; 3Suzhou Institute of Metrology, Suzhou, 215128, China

**Keywords:** silica aerogel, methyltrimethoxysilane, dimethyldimethoxysilane, thermal insulation, radiative cooling, thermal management

## Abstract

Silica aerogels are one of the most widely used aerogels, exhibiting excellent thermal insulation performance and ultralow density. However, owing to their plenitude of Si-O-Si bonds, they possess high infrared emissivity in the range of 8–13 µm and are potentially robust passive radiative cooling (PRC) materials. In this study, the PRC behavior of traditional silica aerogels prepared from methyltrimethoxysilane (MTMS) and dimethyldimethoxysilane (DMDMS) in outdoor environments was investigated. The silica aerogels possessed low thermal conductivity of 0.035 W/m·K and showed excellent thermal insulation performance in room environments. However, sub-ambient cooling of 12 °C was observed on a clear night and sub-ambient cooling of up to 7.5 °C was achieved in the daytime, which indicated that in these cases the silica aerogel became a robust cooling material rather than a thermal insulator owing to its high IR emissivity of 0.932 and high solar reflectance of 0.924. In summary, this study shows the PRC performance of silica aerogels, and the findings guide the utilization of silica aerogels by considering their application environments for achieving optimal thermal management behavior.

## 1. Introduction

Silica aerogels, featuring extremely low density, low thermal conductivity, extremely high porosity, high specific surface area (SSA), and high pore volume, are the most typical aerogels [1,2,3,4,5,6,7,8] since their first discovery in 1931 [9]. Owing to their overall performance such as low thermal conductivity, relatively low cost, and excellent flame resistance, silica aerogels became commercially available in large-scale production and have been potentially used as Cherenkov radiators and drug carriers [3,4,5,10]. Furthermore, they have been widely used for thermal insulation applications due to their extremely low thermal conductivity [11,12,13,14,15,16]. For example, silica-aerogel-based rubber composite has been used as alternative thermal insulation in buildings [11]; a silica-polyimide aerogel blanket has been used as artificial island thermal insulation [12]; a highly elastic silica composite aerogel has been used at extremely low temperatures [13]; silica aerogel phase-change materials have been used for both extremely low and high temperatures [14]; alumina–silica aerogels have been used for thermal insulation at 1500 °C [15]; a highly transparent silica aerogel monolith has been used for solar heating and heat preservation [16]; etc. It can be concluded that almost all the applications of silica aerogels in thermal insulation are based on their low thermal conductivity. 

Silica (SiO_2_) particles, which have the same chemical components (-O-Si-O-) as silica aerogel, show high intrinsic emissivity in the range of 8–13 µm, which coincides with the atmospheric transparency window. Therefore, SiO_2_ particles have been widely used as a functional component to improve emissivity, hence resulting in excellent passive radiative cooling (PRC) performance in polymeric (e.g., polymethylpentene, poly(vinylidene fluoride-co-hexafluoropropene), polylactic acid, and polyethylene) film and textiles [17,18,19,20]. The working principle of PRC is emitting heat into the cold space by radiation in the range of 8–13 µm, in which the heat absorption by the atmosphere is negligible. Therefore, to obtain daytime PRC, high solar reflectance is also required [21,22,23,24,25]. Consequently, silica aerogels must also be excellent PRC materials, and when used in outdoor environments, their PRC ability may outperform their thermal insulation properties. That is, they may be a cooler rather than an insulator to keep warm by heat insulation, as confirmed in our recent work, in which sub-ambient cooling of 10 °C was achieved when silica aerogel particles were used as an additive for a polymer film (thermoplastic polyurethane) [26]. Furthermore, PRC at night was also observed for the traditional transparent silica aerogel: although the silica aerogel was heated in the daytime, its temperature is slightly lower than ambient [16].

To further reveal the PRC behavior of native silica aerogels and solve the challenge of the passive daytime radiative cooling (PDRC) of silica aerogels, in this study, monolithic silica aerogels are synthesized according to Kanamori’s strategy by using methyltrimethoxysilane (MTMS) and dimethyldimethoxysilane (DMDMS) as co-precursors [27,28,29]. The silica aerogel prepared by this method (Figure 1) is white, indicating that it may be a PDRC material. The transparent silica aerogels [30,31,32,33,34] are not considered because they show solar heating in the daytime [16]. The traditional thermal insulation properties of the monolithic silica aerogels will be investigated, and then their passive radiative cooling performance at night and daytime will be studied. Finally, the reasons for the PDRC performance of the silica aerogels will be discussed and the mechanism for the thermal management behavior of the silica aerogel will be proposed.

## 2. Materials and Methods

### 2.1. Materials

MTMS (AR), DMDMS (AR), and ethanol (AR) were obtained from Aladdin Company, Shanghai, China. Acetic acid (99%), cetyltrimethylammonium bromide (CTAB) (99%), and urea (99.5%) were obtained from Sinopharm Chemical Reagent Co., Ltd., Shanghai, China. Distilled water was self-prepared and used as a solvent for silica hydrogels. 

### 2.2. Synthesis of Silica Aerogels

The silica aerogels were synthesized according to the literature [27,28,29]. Firstly, silica precursors, urea, and CTAB were mixed with an aqueous solution of acetic acid (5 mmol/L) in a molar ratio of 1:4:0.1 by stirring. After stirring at room temperature for 1 h, the homogeneous mixture was poured into molds, sealed, and left to stand in an oven at 60 °C for 3 days to complete the gelation and aging. The gels were soaked in deionized water for 2 days and ethanol for another two days to remove residual surfactants and other chemicals. Finally, the silica aerogels were obtained via ambient pressure drying at 100 °C. The volume ratio of the silica precursors was variable; when MTMS:DMDMS was 3:2 and 7:3, the silica aerogels were named MDA3-2 and MDA7-3, respectively.

### 2.3. Characterizations

The solar reflectance spectroscopy of the silica aerogels was tested by a UV-Vis-NIR spectrophotometer (UV-3600PLUS, Shimadzu, Japan) equipped with a gold integrating sphere. The thermal conductivities of the silica aerogels were measured by a 3ω method thermal conductivity measurement device at room temperature, and the measurement results were obtained three times at 5 min intervals between two tests. The surface morphologies of the aerogels were observed by a cold field emission scanning electron microscope (S-4800, Hitachi, Japan) at an acceleration voltage of 20 kV after being sprayed with gold. The water contact angles of the transparent silica aerogels were tested by the video optical contact angle measuring instrument (OCA15EC, Dataphysics, Germany). The infrared emissivity was determined using an FT-IR spectrometer (Bruker INVENIO, Germany) with an integrating sphere (PIKE INTERRATIR). The infrared images of the silica aerogels were taken with an infrared camera (TiX580, Fluke, USA). The temperatures of the samples were measured with thermocouples, which were connected to a temperature acquisition system (JK808, Changzhou Jinailian Co., Ltd., Changzhou, China). The densities of the silica aerogels were calculated by weighing the samples and measuring the volumes. Porosity was calculated according to the equation: porosity = (1 − ρ_b_/ρ_s_) × 100%, where ρ_b_ is the bulk density of the silica aerogel and ρ_s_ is the skeleton density of silica (~2.2 g/cm^3^).

### 2.4. Passive Radiative Cooling Performance of the Silica Aerogels

The radiative cooling performance of the silica aerogels was tested on the roof of a five-story building to ensure full access to the open sky and to exclude thermal radiation from surrounding buildings. The experimental setup consisted of a polystyrene foam box, aluminum foil, low-density polyethylene (LDPE) film, a radiative cooler, high-temperature polyimide tape, thermocouples, and a luminometer [17,18,19,20]. The relative humidity was 40–80%, and the setup was studied under sunlight on sunny and non-cloudy days. The foam box was wrapped with aluminum foil to reduce the temperature of other areas of the foam box owing to heat absorption. A piece of transparent 0.013 mm-thick LDPE film was applied on top of the thermal isolation box to reduce heat convection and conduction between the cavity and the environment. Temperatures were measured using thermocouples placed between the silica aerogel and the substrate, and the thermocouples were in close contact with the aerogels via an adhesive tape, whereas the temperatures of the ambient and black substrate were monitored by the thermocouple suspended in the air and placed on the surface of the black substrate, respectively. Temperature data were stored every 10 s in a USB flash drive using a handheld multichannel thermometer (JK808). Simultaneously, the solar irradiance was recorded by a solar power meter (TES-1333) for daytime PRC.

## 3. Results and Discussion

### 3.1. Synthesis and Characterization of the Silica Aerogels

To study the PDRC performance of silica aerogels, monolith silica aerogels using MTMS and DMDMS as co-precursors were synthesized as presented in Figure 1a [27,28,29,35]. The silica aerogels prepared by these monomers are white and flexible, as shown in Figure 1b–d. When no DMDMS is used, highly transparent silica aerogels can be prepared [30]. However, the solar heating of transparent silica aerogel is dominated by the absence of cooling performance [16,36,37,38]. Interestingly, the introduction of DMDMS can significantly increase the skeleton size of silica aerogels, thus resulting in opaque aerogels due to the scattering of light [39,40,41,42]. In this study, silica aerogels with a weight ratio of MTMS:DMDMS equal to 3:2 and 7:3 were synthesized to study their chemical structures and property relationships. The densities and porosities of the silica aerogels ranged from 0.08 to 0.120 g.cm^3^ and 94.5 to 96.4%, respectively. For convenience, the aerogels were named MDA3-2 and MDA7-3, respectively.

The morphologies of the MDAs are shown in Figure 2a. They are formed by silica particles with neck structures. The silica particles are as large as 2–5 µm in diameter and they possess relatively small SSA [27,28], thus the SSA of the MDA was not determined by the BET method. Large pores of up to tens of micrometers are formed between the silica particles. All the structural characteristics offered the unobservable shrinkage of the silica aerogel by ambient pressure dying. The energy-dispersive X-ray spectrometry (EDX) mapping shown in Figure 2b,c further confirms the silica particle structures. The main elements of SiO_2_, O, and Si can be clearly observed, and they also show particle shapes. Furthermore, C elements can also be observed, which are the -CH_3_ groups from MTMS and DMDMS. The relative C weight ratio (which excludes the content of light element H) in MDA3-2 and MDA7-3 is 5.52 wt.% and 18.71 wt.%, respectively (Appendix A). The results suggest that increasing the DMDMS can increase the content of -CH_3_ groups in the silica aerogels because there are two -CH_3_ groups in each DMDMS monomer and only one -CH_3_ group in MTMS. Owing to the plenitude of -CH_3_ groups in the MDAs, they are hydrophobic with contact angles higher than 110° (Figure 2d), and water can form droplets on the surface (Figure 2e). Nevertheless, the thermal conductivities of the MDAs are low, in the range of 0.035~0.045 W/m·K (Figure 2f). Although the values are relatively higher than that of silica aerogels prepared with other monomers and supercritical drying [16,27,28,29,43], they are comparable to silica aerogels prepared from water glass by ambient pressure drying [14] and should be good thermal insulators, as will be confirmed in the next section.

### 3.2. Thermal Insulation Behavior of the Silica Aerogel

The thermal insulation behavior of the silica aerogels is illustrated in Figure 3. As shown in Figure 3a, silica aerogels, MDA3-2, with a thickness of 1.5 and 4.5 mm were presented on a hot stage test. Their thermal insulation capacity is illustrated in Figure 3b. The IR photos of the MDA3-2 presented on a hot stage with different temperatures were taken with an IR camera after reaching a constant temperature (Figure 3c). It can be seen that the top surface temperatures are significantly lower than the hot stage. Figure 3d–g show the temperature changes vs time of the MDA on the hot stage with temperatures of 100, 200, 300, and 400 °C, respectively. The top surface temperatures increased and reached steady states within 100 to 550 s, depending on thse temperature of the hot stage. The specific temperature differences (ΔT) of the two aerogels with different thicknesses were 43.7 and 28.9 °C when the hot stage was 100 °C. Impressively, the ΔT of 184 °C was achieved with a thickness of 1.5 mm when the hot stage was 400 °C, and ΔT remarkably increased to 232.5 °C when the thickness was 4.5 mm. The results suggest that the MDA possesses good thermal insulation capacities in a wide range of temperatures.

### 3.3. PDRC Performance of the Silica Aerogels

Figure 4a,b show the setup used to evaluate the PRC performance of the silica aerogels in the outdoor environment at night. In the experiment, the silica aerogels were used as passive radiative coolers. A thermocouple was suspended in the air to measure the ambient temperature. The PRC results are shown in Figure 4c. The ambient temperature gradually decreased from 9 °C at 20:00 to 5 °C the next morning at 6:00. Notably, regardless of the low thermal conductivity and sound thermal insulation property of the silica aerogels, the temperatures of the silica aerogel were lower than the ambient; sub-ambient cooling of 12 and 10 °C was achieved at 6:00 for MDA7-3 and MDA3-2, respectively. Interestingly, the white, opaque silica aerogels also show the same passive cooling capacity as the transparent silica aerogels. Moreover, the cooling is even more robust for the MDA, and the cooling temperature can be slightly varied by the volume ratio of MTMS to DMDMS. 

In fact, for most of the materials with relatively high IR emissivity, passive cooling can be achieved on clear nights but it is a challenge to cool in the daytime owing to the robust solar heating effect [44,45]. Therefore, high solar reflectance is required to reach effective daytime radiative cooling. To investigate the PDRC performance of the silica aerogels, Figure 5a,b show the setup used to evaluate the PDRC performance in the outdoor environment. In the experiment, the black base and cavity temperatures of the equipment were measured. Thermocouples were placed at the bottom of each sample and on the surface of the black base. A thermocouple was suspended in the cavity to measure the cavity temperature. Considering the characteristics of the solar radiation intensity on 14 December 2022, the time for the experiments was chosen to be from 13:00 to 15:00. The PDRC results are shown in Figure 5c. The solar irradiance is weak in winter, ranging from 400 to 500 W·m^−2^. Nevertheless, solar heating is still robust under relatively low irradiation, the temperature of the black base could be heated up to 40 °C at 13:00, and it was 20 °C at 15:00 when the cavity temperature dropped to 10 °C. Notably, the temperatures of the silica aerogels were lower than that of the cavity. The average cooling temperatures for the MDA3-2 and MDA7-3 were 5.5 and 7.5 °C, respectively. In summary, an obvious daytime cooling was observed in the MTMS/DMDMS-based silica aerogels, which we can attribute to their high solar reflectance and high IR emissivity. 

### 3.4. Reasons and Proposed Mechanism for PDRC

Figure 6a shows the spectral reflectance of the silica aerogels. The average reflectance values of MDA3-2 and MDA7-3 were 0.858, and 0.924, respectively. The results confirmed that the introduction of DMDMS can significantly increase the solar reflectance of the silica aerogels owing to their large pore size and silica particles. sThe results indicated that solar reflectance is the critical parameter to fulfill PDRC, as has also been confirmed in the literature for other radiative cooling materials [45,46,47,48,49,50]. Moreover, the IR emissivity spectra shown in Figure 6b indicated that both MDA7-3 and MDA3-2 possessed high emissivity in the range of 8–13 µm. The average emissivity values of MDA7-3 and MDA3-2 were 0.932 and 0.946, respectively. The high emissivity of the silica aerogel may be due to the plenitude of Si-O-Si and Si-C bonds, whose fingersprint area of the silica aerogel ranged from 1300 to 600 cm^−1^ in the Fourier transform infrared images (Appendix A), coinciding with the atmospheric transparency window (8–13 µm). The strong and highly selective absorption of the silica aerogels can significantly contribute to the high emissivity of the silica aerogel and further contribute to the powerful PDRC performance of the silica aerogels.

Energy exchange between materials and the environment includes thermal conduction and convection, solar heating, outward thermal radiation, and thermal radiation from the environment. Under the condition of thermal equilibrium, the net power of gain *P_net_* [44,45] can be referred to as Equation (1):(1)$$p_{net}\left(T\right)=p_{rad}\left(T\right)-p_{atm}\left(T_{amb}\right)-p_{solar}-p_{conv+cond}$$
where *P_rad_* represents the energy that the sample emits outward, *P_solar_* is the radiation received from the sun, *P_atm_* is the radiation from the atmosphere, and *P_cond+conv_* is the non-radiative heat exchange with the environment through thermal conduction and convection.

In traditional applications, the thermal management of silica aerogels is only considered with their low thermal conductivities. However, ins specific conditions, such as outdoor use, the silica aerogel interacts with the entire environment, and its net energy gain must be referred to as Equation (1) rather than just considered with thermal conductivity. Therefore, the thermal management behavior of the MTMS–DMDMS-based silica aerogel is proposed in Figure 7. The heat gain from the sun can be reflected and emitted strongly to space, which affords the silica aerogel with robust PDRC capacity. Thus, when silica aerogels are used for thermal insulation directly in the outdoor environment, their PDRC performance should be considered to obtain ideal thermal management results. It is worth noting that when silica aerogel is designed for thermal insulation purposes, silica aerogels with MTMS as the single component may be preferable owing to the reduced PDRC and lower thermal conductivity. 

## 4. Conclusions

In summary, silica aerogels with low thermal conductivity, high solar reflectance, and high IR emissivity are synthesized by using MTMS and DMDMS as co-precursors. The silica aerogels exhibit good thermal insulation on a hot stage in the room environment, which is similar to traditional aerogels. Interestingly, the robust PDRC performance of the silica aerogels was confirmed; they showed an impressive sub-ambient cooling of 7.5 °C in the daytime and 12 °C at night in an outdoor environment. The excellent PDRC of silica aerogels can be attributed to their high solar reflectance and IR emissivity. The results indicated when silica aerogels are used for thermal insulation, negative results may be obtained because they are robust coolers. The findings in this study provide new insight into silica aerogels and help to guide comprehensive consideration when using aerogels for thermal management in an outdoor environment. 

## Figures and Tables

**Figure 1 nanomaterials-13-00467-f001:**
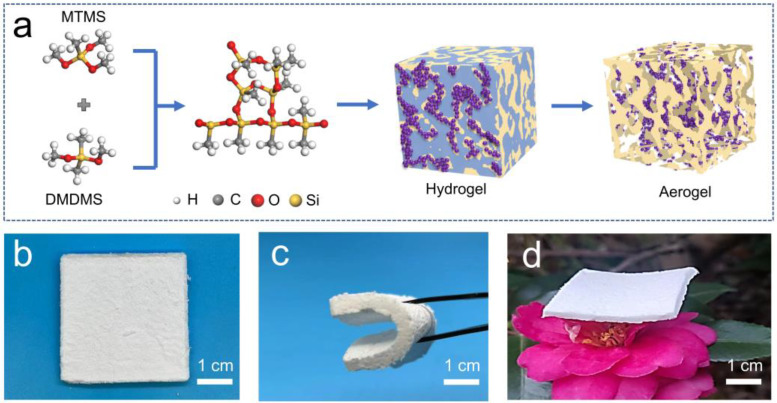
(**a**) Illustration of the synthetic approach to the MTMS–DMDMS aerogel (MDA); (**b**–**d**) photo images of an MDA, a blended DMA, and an MDA standing on a flower, respectively.

**Figure 2 nanomaterials-13-00467-f002:**
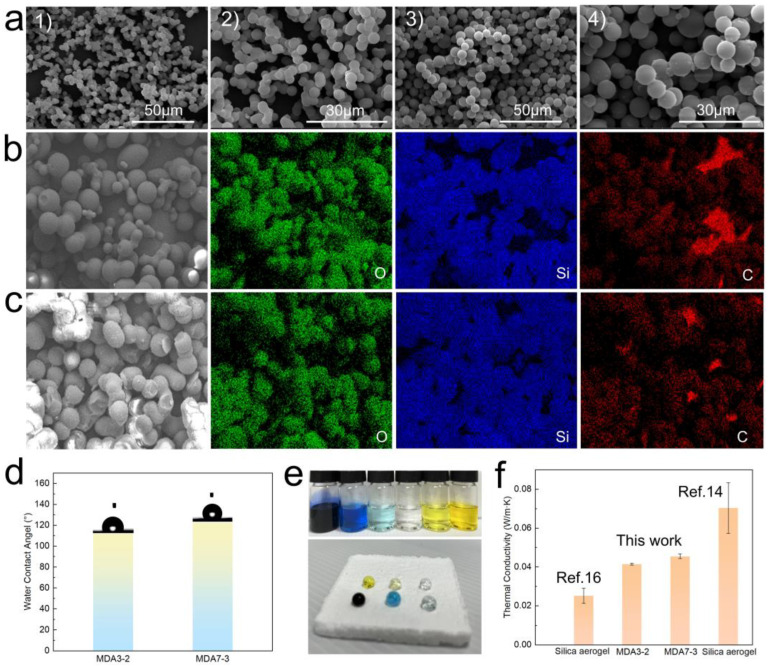
(**a**) SEM images of the MDAs: *1* and *2* are MDA3-2, *3* and *4* are MDA7-3; (**b**) EDX mapping of MDA3-2: the O, Si, and C elements are highlighted in green, blue, and red, respectively; (**c**) EDX mapping of MDA7-3: the O, Si, and C elements are highlighted in green, blue, and red, respectively; (**d**) contact angles of the MDAs; (**e**) photo images of colored water and their droplets on MDA; (**f**) thermal conductivity of the MDAs.

**Figure 3 nanomaterials-13-00467-f003:**
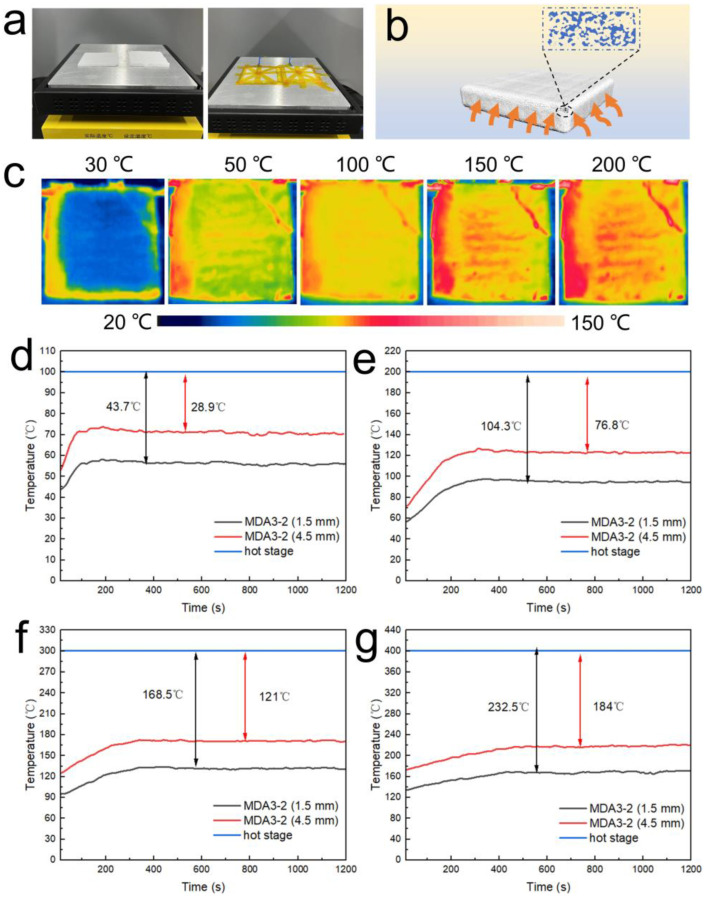
(**a**) Photo images of the MDAs presented on a hot stage; (**b**) carton image illustrating the thermal insulation performance of MDA; (**c**) IR images of the MDAs presented on the hot stage; (**d**) the temperature changes of the upper side of the MDA on a hot stage of 100 °C; (**e**) the temperature changes of the upper side of the MDA on a hot stage of 200 °C; (**f**) the temperature changes of the upper side of the MDA on a hot stage of 300 °C; (**g**) the temperature changes of the upper side of the MDA on a hot stage of 400 °C.

**Figure 4 nanomaterials-13-00467-f004:**
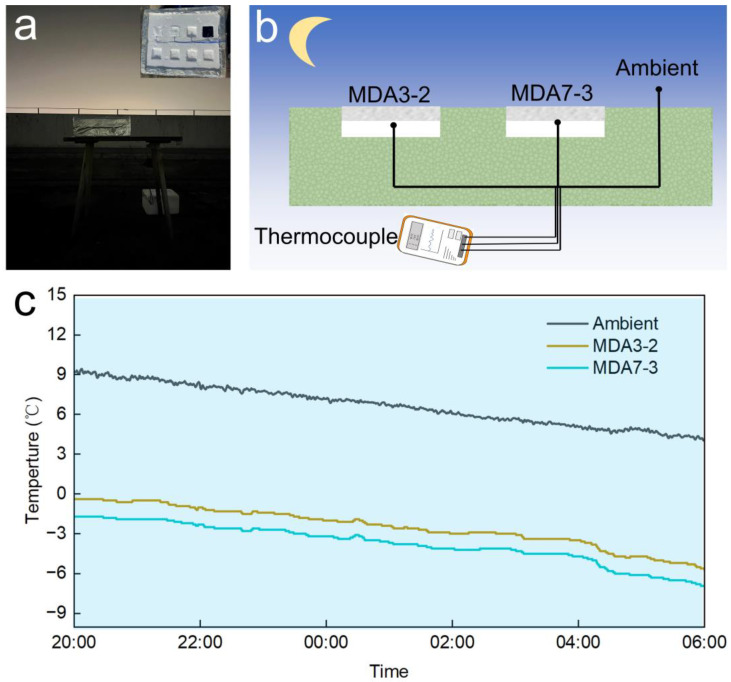
(**a**) Photo image of PRC measurement at night in Suzhou, 12–13 December 2022; (**b**) schematic image of the system used to characterize the PRC; (**c**) temperature tracking of the aerogel and ambient temperature during PRC.

**Figure 5 nanomaterials-13-00467-f005:**
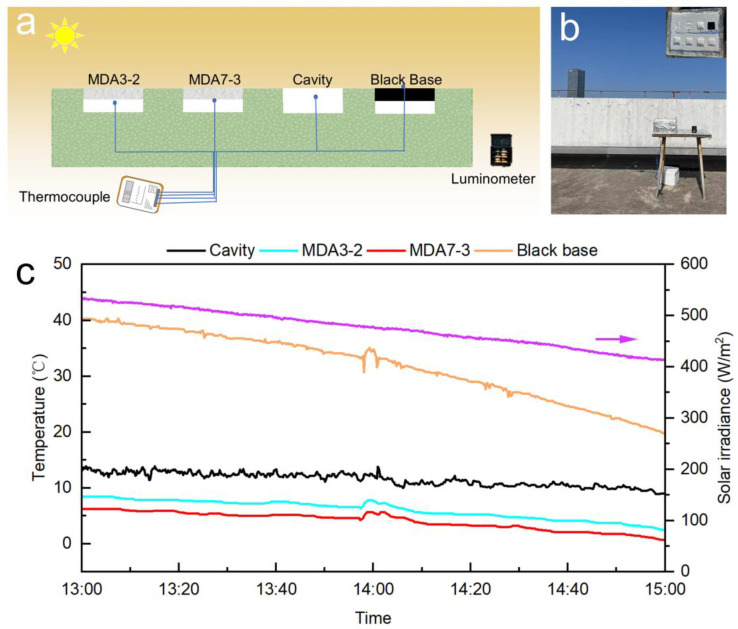
(**a**) Schematic of the setup used to evaluate the radiative cooling performance; (**b**) photo image of the setup used to evaluate the radiative cooling performance in Suzhou, 14 December 2022; (**c**) temperature tracking of the aerogel, black base, and ambient. The monitored solar irradiance is included to provide primary meteorological information.

**Figure 6 nanomaterials-13-00467-f006:**
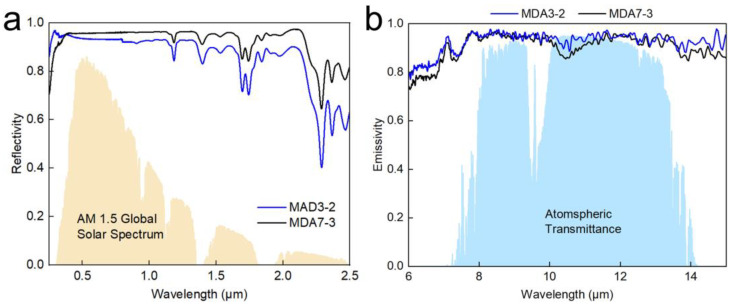
(**a**) Solar reflective spectra of the aerogels. The normalized ASTM G173 global solar spectrum is plotted in the background; (**b**) thermal emissive spectra of the aerogels. Atmospheric transparency windows are plotted as background.

**Figure 7 nanomaterials-13-00467-f007:**
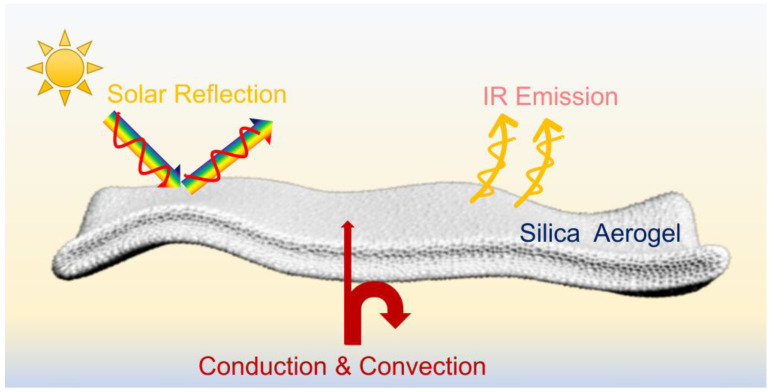
Proposed thermal management behavior of silica aerogels with the combination of thermal insulation and passive radiative cooling.

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
