# Peer review of "Passive Daytime Radiative Cooling of Silica Aerogels"

_nanomaterials, 2023, doi:10.3390/nano13030467_

Round 1

Reviewer 1 Report

The authors prepared hydrophobic silica aerogels in this manuscript and investigated their thermal and passive radiative cooling (PRC) properties. The authors proposed a mechanism that better explains the silica aerogels' thermal management behavior. The manuscript needs to be improved before accepting for publication.

My comments and suggestions are listed below

1.       Line 61: authors refer to Scheme 1, which is missing in the text. I guess it is Fig. 1a

2.       Line 69: ensure you list all the chemicals and their origin in the Materials section.

3.       Line 80: ethanol, maybe

4.       Line 131: caption in fig.1 can be improved by reducing the repetition

5.       Line 137: do you think the SiO2 particles themselves might be porous? And BET surface area might be high due to meso- and micropores? If the material consists of only macropores, is the aerogel's name reliable here? Maybe other terms, like ambigel, are better suited?

6.       Line 138-142: I think EDX mapping results are not carrying important information here. Furthermore, light elements, like carbon, cannot be quantitatively determined by EDX. For carbon content, you might need to do CHN elemental analysis. Or TGA.

7.       Lines 148-150: what do you mean by "traditional" and "other" aerogels?

8.       Fig 2f: a column with the thermal conductivity of a reference insulating material or from literature makes the data look better

9.       Lines 171-172: are these values higher/lower, and how, compared to results achieved before in the literature?

10.   Fig. 5c: better color code for the curves is needed

11.   Line 211: how could you explain the difference in cooling temperature for the samples MDA3-2 and MDA7-3 from a compositional and microstructure point of view?

12.   Line 212: What do you mean by "most famous" silica aerogel?

13.   Line 231: Is there any influence of Si-C or C-C bonds in the emissivity of these aerogels?

14.   The overall language of the manuscript needs to be more scientific.

Author Response

The authors prepared hydrophobic silica aerogels in this manuscript and investigated their thermal and passive radiative cooling (PRC) properties. The authors proposed a mechanism that better explains the silica aerogels' thermal management behavior. The manuscript needs to be improved before accepting for publication.

Reply: Thanks for your careful review and kind suggestions. I have revised the manuscript according to your suggestions.

1. Line 61: authors refer to Scheme 1, which is missing in the text. I guess it is Fig. 1a

Reply: Scheme 1 has been revised to Figure 1 in the revised manuscript.

2. Line 69: ensure you list all the chemicals and their origin in the Materials section.

Reply: The purities of acetic acid, CTAB, and urea are provided in the revised manuscript, and the information on distilled water and ethanol was provided in the materials section, as follows:

“MTMS (AR), DMDMS (AR), and ethanol (AR) were obtained from Aladdin Company, Shanghai, China. Acetic acid (99%), cetyltrimethylammonium bromide (CTAB) (99%), and urea (99.5%) were obtained from Sinopharm Chemical Reagent Co., Ltd., Shanghai, China. Distilled water was self-prepared and used as a solvent for silica hydrogels.”

3. Line 80: ethanol, maybe

Reply: Thanks for your careful review, the word ethanal has been revised to ethanol.

5. Line 131: caption in fig.1 can be improved by reducing the repetition

Reply: The caption for Fig. 1 has been revised to

“Figure 1. (a) Illustration of the synthetic approach of the MTMS-DMDMS aerogel (MDA); (b)-(d) Photo images of an MDA, a blended DMA, and an MDA stands on a flower, respectively.”

5. Line 137: do you think the SiO2 particles themselves might be porous? And BET surface area might be high due to meso- and micropores? If the material consists of only macropores, is the aerogel's name reliable here? Maybe other terms, like ambigel, are better suited?

Reply: Thanks for the kind suggestions. When Kanamori et al. first synthesized the MTMS-derived aerogels, they indeed named them “marshmallow-like” aerogels and xerogels because they were obtained by ambient pressure drying (ref. 27, 28, 29, and 30). However, during the past decade, ambient pressure drying became one of the most widely used drying methods to prepare silica aerogels, thus many scientists used aerogel to name the xerogel prepared by ambient pressure drying. Similar works published on Micropor. Mesopor. Mater. 2012, 148, 145-151(ref. 42) also used aerogels. Thus, even though there were no micropores and mesopores in the silica particles of this study, they were still widely called “aerogels”. One of the most important reasons is that they were prepared by the sol-gel transition, and underwent a drying process with limited shrinkage, the structure before and after drying did not have significantly changed. Besides, the aerogel prepared by the ambient pressure drying also possessed low density (0.080~0.120 g/cm3), low thermal conductivity (0.035 ~ 0.045 W/m·K), and high porosity.

6. Line 138-142: I think EDX mapping results are not carrying important information here. Furthermore, light elements, like carbon, cannot be quantitatively determined by EDX. For carbon content, you might need to do CHN elemental analysis. Or TGA.

Reply: Thanks for the kind reminder. Yes, the element content by EDX is not quantitatively, in this work, the carbon content is relatively compared. We have revised the manuscript by pointing out this issue, as follows:

“The energy-dispersive X-ray spectrometry (EDX) mapping shown in Figures 2b and 2c further confirmed the silica particle structures. The main elements of SiO2, O and Si can be clearly observed, and they also showed particle shapes. Furthermore, C elements can also be observed, which are the -CH3 groups from MTMS and DMDMS. The relative C weight ratio (which excluded the content of light element H) in MDA3-2 and MDA7-3 is 5.52 wt.% and 18.71 wt.%, respectively (Figure S1 and S2 in the supplementary information).”

7. Lines 148-150: what do you mean by "traditional" and "other" aerogels?

Reply: To be more clearly presented, the description in the revised manuscript has been revised by mentioning the aerogels described in the cited references:

“Although the values are relatively higher than that of traditional silica aerogels prepared with other monomers and supercritical drying [16, 27-29, 43], they are comparable to other silica aerogels prepared from water glass by ambient pressure drying [14] and should be good thermal insulators and will be confirmed in the next section.”

8. Fig 2f: a column with the thermal conductivity of a reference insulating material or from literature makes the data look better.

Reply: Reference insulating materials from ref. 14 and 16 are added to Fig. 2f, as follows:

(please see the Fig in the attached word files)

9. Lines 171-172: are these values higher/lower, and how, compared to results achieved before in the literature?

Reply: Because there is no standard method to evaluate the actual thermal insulation performance of aerogels, for instance, the target temperatures, the thickness of the aerogels, and the ambient temperatures are varied from Lab to Lab, therefore the thermal insulation performance of the silica aerogel presented in lines 171-172 did not compared to literature. However, the potential thermal insulation capacity is comparable by thermal conductivity. Consequently, the thermal conductivity values in Fig. 2f has been revised as suggested.

10. Fig. 5c: better color code for the curves is needed

Reply: Fig. 5c has been revised as suggested. Higher contrast color code for each curves are provided.

(please see the Fig in the attached word files)

11. Line 211: how could you explain the difference in cooling temperature for the samples MDA3-2 and MDA7-3 from a compositional and microstructure point of view?

Reply: Thanks for the reminds. To explain the cooling temperatures, more data of their solar reflectance and IR emissivity values are needed. Therefore, in line 211 we only presented the experimental results. The reasons to explain the difference are provided in section 3.4, the IR emissivity is compositionally related due to the Si-O-Si bonds, and solar reflectance is related to the large size of silica particles.

“Figure 6a shows the spectral reflectance of the silica aerogels. The average reflectance values of MDA3-2 and MDA7-3 were 0.858, and 0.924, respectively. The results con-firmed that the introduction of DMDMS can significantly increase the solar reflectance of the silica aerogels owing to their large pore size and silica particles. It can be seen from Figure 5c that the cooling performance of MDA7-3 is stronger than MDA3-2, which is can be explainable due to the higher reflectance of MDA7-3. The results indicated that solar reflectance is the critical parameter to fulfill PDRC, as has also been confirmed in literature for other radiative cooling materials [45-50]. Moreover, the IR emissivity spectra shown in Figure 6b indicated that both MDA7-3 and MDA3-2 possessed high emissivity in the range of 8 – 13 µm. The average emissivity values of MDA7-3 and MDA3-2 were 0.932 and 0.946, respectively. The high emissivity of the silica aerogel may be due to the plenty of Si-O-Si bonds, whose fingerprint area of the silica aerogel ranged from 1300 to 600 cm-1 in the Fourier transform infrared (FT-IR) coincides with the atmospheric transparency window (8-13 µm). The strong and highly selective absorption of the silica aerogels can significantly contribute to the high emissivity of the silica aerogel, and further contribute to the powerful PDRC performance of the silica aerogels.”

12. Line 212: What do you mean by "most famous" silica aerogel?

Reply: We feel sorry to use the unclear description. The problem has been revised as follows:

“In sum, an obvious daytime cooling was observed in the MTMS/DMDMS based silica aerogels, which we can attribute to their high solar reflectance and high IR emissivity.”

13. Line 231: Is there any influence of Si-C or C-C bonds in the emissivity of these aerogels?

Reply: There were Si-CH3 groups in the silica aerogels from MTMS and DMDMS, the Si-C bonds were located at 840 cm-1, which is also in the range of the atmospheric transparency window. Thus they can contribute to the emissivity, even if the content of Si-C is lower than that of Si-O-Si bonds. In the revised manuscript, the influence of Si-C bonds in the emissivity is added:

“the average emissivity values of MDA7-3 and MDA3-2 were 0.932 and 0.946, respectively. The high emissivity of the silica aerogel may be due to the plenty of Si-O-Si and Si-C bonds, whose fingerprint area of the silica aerogel ranged from 1300 to 600 cm-1 in the Fourier transform infrared (Figure S3) coincides with the atmospheric transparency window (8-13 µm).”

Figure S3 is provided in the revised supporting information:

(please see the Fig in the attached word files)

14. The overall language of the manuscript needs to be more scientific.

Reply: After the careful reversions according to the constructive suggestions of 1 to 13, as well as the reversions by the authors, we hope the language of the revised manuscript is more scientific. All the other changes can be found with a “Track Changes” function.

Reviewer 2 Report

- More informations about the physical values are expected, e.g. how was the porosity or density od obtained aerogels

- Authors use the ambient pressure drying for obtained the aerogels, probabely because of the relatively Å‚ów cost of this method. However, it is well-known from literature that this kind of drying (in opposite to supercritical drying) may lead to high values of volume shrinkage. Did the authors analyzed this problem and could they add a few words of comment in this subject?

- Authors investigeted two weight ratios MTMS:DMDMS. Both arised to have good insulating properties. However, MDA7-3 seems to have a little bit better properies. This should be commented/discussed somwhere in manuscript.

- Following the previous remark: the sample MDA7-3 has better properties than MDA3-2; also the first sample contains (by per cent) more MTMS than the second. Thus, can we conclude that the addition of DMDMS worsens  our insulation material? May be it would be better to synthetise the aerogel basing on MTMS only? Discussion needed.

- BTW: in first line of caption to Fig. 2 please correct MDA7-2 to MDA7-3.

Author Response

Dear reviewers, thanks a lot for your careful review and constructive suggestions, which can remarkably increase the quality of this study. Following is our point-to-point response to your suggestions.

- More informations about the physical values are expected, e.g. how was the porosity or density of obtained aerogels

Reply: the information has been added in the revised manuscript:

“In this study, silica aerogels with a weight ratio of MTMS:DMDMS equal to 3:2 and 7:3 were synthesized to study their chemical structure and property relationship. The densities and porosities of the silica aerogels ranged from 0.08 to 0.120 g.cm3 and 94.5 to 96.4%, respectively. For convenience, the aerogels are named MDA3-2 and MDA7-3 respectively.”

- Authors use the ambient pressure drying for obtained the aerogels, probably because of the relatively low cost of this method. However, it is well-known from literature that this kind of drying (in opposite to supercritical drying) may lead to high values of volume shrinkage. Did the authors analyzed this problem and could they add a few words of comment in this subject?

Reply: Thanks for the constructive suggestions. As has been mentioned in the manuscript, the introduction of co-monomer DMDMS resulted in large pore size, thus no obvious shrinkage during ambient pressure drying was observed. Therefore, we added a few comments on this subject in the revised manuscript:

“The morphologies of the MDA are shown in Figure 2a. They are formed by silica particles with neck structures. The silica particles are as large as 2-5 µm in diameter and they possess relatively small SSA [27,28], thus the SSA of the MDA was not determined by the BET method. Large pores up to tens of micrometers are formed between the silica particles. All the structural characters offered unobservable shrinkage of the silica aerogel by the ambient pressure dying.”

- Authors investigated two weight ratios MTMS:DMDMS. Both arised to have good insulating properties. However, MDA7-3 seems to have a little bit better properties. This should be commented/discussed somewhere in manuscript.

Reply: Thanks for the meaningful suggestions. In section 3.4 Reasons and proposed mechanism for PDRC, comments and discussion of the better PDRC property of MDA7-3 have been added:

“The results confirmed that the introduction of DMDMS can significantly increase the solar reflectance of the silica aerogels owing to their large pore size and silica particles. It can be seen from Figure 5c that the cooling performance of MDA7-3 is stronger than MDA3-2, which is can be explainable due to the higher reflectance of MDA7-3. The results indicated that solar reflectance is the critical parameter to fulfill PDRC, as has also been confirmed in literature for other radiative cooling materials [45-50].”

- Following the previous remark: the sample MDA7-3 has better properties than MDA3-2; also the first sample contains (by percent) more MTMS than the second. Thus, can we conclude that the addition of DMDMS worsens our insulation material? Maybe it would be better to synthesize the aerogel based on MTMS only? Discussion needed.

Reply: The introduction of DMDMS results in large silica particle size and pore sizes, which increase its thermal conductivity. Therefore, when silica aerogels are used only for thermal insulation, it is indeed the addition of DMDMS worsens the insulation materials, and it’s better to synthesize aerogel based on MTMT only (see ref. 16). However, the novelty of this study is PDRC, which is used for daytime cooling purpose, thus, the addition of DMDMS can fulfill the radiative cooling target better than MTMS. Discussion is provided in the last sentence in section 3.4:

“Therefore, the thermal management behavior of the MTMS-DMDMS-based silica aerogel is proposed in Figure 7. The heat gain from the sun can be reflected, and emitted strongly to space, which affords the silica aerogel with robust PDRC capacity. Thus, when silica aerogels were used for thermal insulation directly in the outdoor environment, their PDRC performance should be considered to obtain ideal thermal management results. It is worth noting that when silica aerogel is designed for thermal insulation purposes, silica aerogels with MTMS as the singlet component may be preferable owing to the reduced PDRC and lower thermal conductivities.”

- BTW: in first line of caption to Fig. 2 please correct MDA7-2 to MDA7-3.

Reply: DMA7-2 has been corrected to MDA7-3.

Reviewer 3 Report

The article leaves a good impression. The novelty of the material is at a good level.

There are several comments.

1. The introduction lacks information about what materials can potentially be used in Passive daytime radiative cooling except for aerogels. In addition, there are other directions for the use of transparent aerogels as detectors of Cherenkov radiation.

2. There are questions about the specific surface of the samples, how does the specific surface affect PDRC performance. For aerogels, the specific surface area is one of the important parameters.

3. And the last question. How resistant are aerogels based on methyltrimethoxysilane (MTMS) and dimethyldimethoxysilane (DMDMS) to sunlight?

Author Response

The article leaves a good impression. The novelty of the material is at a good level.

Reply: Thanks for your positive comments on the manuscript, we have revised the manuscript according to your meaningful suggestions.

1. The introduction lacks information about what materials can potentially be used in Passive daytime radiative cooling except for aerogels. In addition, there are other directions for the use of transparent aerogels as detectors of Cherenkov radiation.

Reply: The information of other materials (such as polymer film and textile) can be potentially used in PDRC is provided in the revised manuscript, textile and film prepared by various polymers with high IR transparency and high IR emissivity can be used for PDRC:

“Therefore, SiO2 particles have been widely used as a functional component to improve emissivity, hence resulting in excellent passive radiative cooling (PRC) performance in polymeric (e.g. polymethylpentene, poly(vinylidene fluoride-co-hexafluoropropene), polylactic acid, and polyethylene) film and textiles [17-20].” And “That is, they may be a cooler rather than an insulator to keep warm by heat insulation, as confirmed in our recent work, in which sub-ambient cooling of 10 ºC was achieved when silica aerogel particle was used as addictive for a polymer film (thermoplastic polyurethane) [26].”

 In the introduction, the transparent aerogels as detectors of Cherenkov radiation, as well as their application as drug carriers and catalyst supports, were not introduced because such applications do not relate to their thermal insulation applications. Nevertheless, as potential Cherenkov radiation application are provided in the revised manuscript:

“Owing to their overall performance such as low thermal conductivity, relative relatively low cost, and excellent flame resistance, silica aerogels became commercially available in large production and been potentially used as Cherenkov radiator and drug carriers [3-5,10]. Besides, they have been widely used for thermal insulation application applications due to their extremely low thermal conductivities [11-16]. For example, silica aerogel-based rubber composite has been used as alternative buildings’ thermal insulation [11]….”

2. There are questions about the specific surface of the samples, how does the specific surface affect PDRC performance. For aerogels, the specific surface area is one of the important parameters.

Reply: Because the introduction of DMDMS, the building block of the silica aerogel is as large as 2-5 µm in diameter, they possess relatively small SSA [also reported in refs. 27,28, and 42]. Moreover, the PDRC performance is many due to high solar reflectance resulted from large pores, so the SSA is not considered in the PDRC performance. For instance, ref. 44 (Hierarchically porous polymer coatings for highly efficient passive daytime radiative cooling. Science 2018, 362, 315) reported Hierarchically porous structure for polymer film, the reflectance is more maybe more related with pore sizes and porosity. Therefore, based on the two reasons, SSA was not studied in this work. Nevertheless, we had discussed the reason of PDRC in section 3.4 by mentioning their microstructures and solar reflectances.

3. And the last question. How resistant are aerogels based on methyltrimethoxysilane (MTMS) and dimethyldimethoxysilane (DMDMS) to sunlight?

Reply: Thanks for the constructive question. For PDRC, resistant to sunlight is an important issue, our previous work by using PU polymer is suffered from poor resistant to sunlight (ref. 26 Adv. Sci. 2022, 9, 2201190). However, the silica aerogels, which is many composed Si-O-Si bonds, and little amount of Si-C bonds, which are chemically inert to most chemical solvents and UV light, are much more stable than the PU polymer and PE. That is why in this work we used all silica derived monomers.

Round 2

Reviewer 1 Report

The manuscript looks good to me and can be accepted for publication in Nanomaterials.